

# A Revised Mid-Pliocene Composite Section for ODP Site 846

√Timothy D. Herbert[1], *Rocio Caballero-Gill[1,2] and Joseph B. Novak[1]

[1]Department of Earth, Environmental and Planetary Sciences, Box 1846, Brown University,
Providence RI 02906 U.S.A.
[2]Atmospheric, Oceanic, and Earth Sciences, George Mason University, University Dr., Fairfax
VA 22030 USA

*Correspondence to*: Timothy D. Herbert (timothy_herbert@brown.edu)

Manuscript date: October 21, 2019

## Abstract

The composite section from ODP Site 846 has provided key data sets for Pliocene stable isotope
and paleoclimatic time series.  We document here errors in primary data sets for stable isotopes
and alkenone-derived sea surface temperature estimates (SST) in the late Pliocene interval
containing the M2 glaciation (ca. 3.290-3.3 Ma) by tying high resolution core measurements to a
continuous downhole conductivity log.  In addition, we provide new stable isotopic and alkenone
measurements that correlate well to the revised splices of color reflectance and gamma ray
attenuation porosity evaluator data from Site 846, and to a new composite section produced at
equatorial Pacific ODP Site 850. A new composite splice for Site 846 is proposed, along with
composite isotope and alkenone time series that should be integrated into revised Pliocene
paleoclimatic stacks.

**Keywords:** Pliocene paleoceanography, stable isotopes, alkenone SST, Pliocene stratigraphy,
hydraulic piston coring




## 1 Introduction

Continuous proxy time series from ODP Site 846 occupy a privileged place in Pliocene
stratigraphy and paleoclimatology. Drilled during Leg 138, this site was one of the first to
systematically integrate high resolution, non-destructive core scanning into the construction of
composite sections (Shipboard Scientific Party, 1992; Hagelberg et al., 1995) that formed the
basis for later detailed shore-based studies. The creation of composite sections was necessitated
by the fact that coring does not completely recover the sequence at any one drill hole; continuous
records must be stitched together to fill in coring gaps. Among the fruits of this work were some
of the longest, best resolved time series of Pliocene stable isotopic variations (Shackleton et al.,
1995b), lithological variations (Hagelberg et al., 1995) and tropical sea surface temperatures
(Lawrence et al., 2006; Herbert et al., 2010) and a proposal for an orbitally-tuned late Neogene
time scale (Shackleton, 1995a). The continuity and high sedimentation rate at site 846 resulted
in its isotopic data set playing an outsized role in the Pliocene interval of the widely used LR04
isotopic stack (Lisiecki and Raymo, 2005. However, it was noted (Figure 9 of Lisiecki and
Raymo, 2005) that the isotopic record in the interval surrounding the glacial event M2 (circa 3.3
Ma) at Site 846 is anomalous in comparison to other sites.

In this contribution we document significant errors in the late Pliocene (equivalent to the Gauss
magnetochron) composite section at Site 846. These affect the stable isotopic and
paleotemperature reconstructions around the Pliocene M2 isotopic event, the most notable
Pliocene glacial stage preceding cyclic northern hemisphere glaciation at ~ 2.7 Ma (Mudelsee
and Raymo, 2005), and immediately predating the well-studied PRISM interval (Dowsett, 2007).
We were drawn to revisit the original composite section because features surrounding the M2
glacial event seemed anomalous in comparison to other time-equivalent marine sections we have
been investigating. We construct a more reliable composite section through a combination of
tying high resolution measurements made at offset holes to a high-resolution downhole log
acquired at Hole 846B, and we supplement that stratigraphic analysis with new stable isotopic
and alkenone sea surface temperature (SST) estimates from hole 846B, which was not used in
the original composite in the M2 interval. We propose a revised composite section spanning the
interval equivalent to the early Gauss through Mammoth magnetic polarity zones (circa 3.2-3.6.
Ma) and tie the ODP Site 846 records to ODP Site 850, which provides an indirect tie of stable
isotopic and SST estimates at Site 846 to a high-quality magnetic polarity stratigraphy. Our
results also yield insights into core recovery and coring distortion artifacts during hydraulic
piston coring of calcareous-siliceous sediments of the equatorial Pacific.

## 2 Methods

### 1.1 Composite section generation


Assignments of gaps in core recovery and distortions induced by coring create challenges in
accurately compositing sedimentary records from offset holes. The scientific part on board Leg
138 made excellent use of high-resolution measurements from the Gamma Ray Porosity
Evaluation scanner, a reflectance spectrophotometer, and magnetic susceptibility logging of



cores (Hagelberg et al., 1995; Mix et al., 1995). Constructing a composite section proceeded by identifying tie points between holes on a core-by-core basis, splicing missing material at core breaks by the section represented at an offset hole, and assuming that a linear offset applied between the drilling depth (mbsf) and the composite depth that resulted from splicing materials between holes.  As compared to the original drilling depth, the composite section at Site 846

grew by about 15%, indicating that a significant amount of material was not recovered at any one hole during drilling.

Later sampling followed the logic of the initial splicing, hopping between holes to follow the composite section documented in the Initial Reports volume (Hagelberg et al., 1992, 1995).  A

more flexible and robust procedure for composite section generation was subsequently developed using dynamic programming to optimally align time series data (Lisiecki and Lisiecki, 2002, and Lisiecki and Herbert, 2007.  This method allowed for assessing distortions in the coring process (e.g. stretching or squeezing of sedimentation relative to the undisturbed section) but it still required user guidance of key tie points and the more or less arbitrary assumption of

which section of several replicates (e.g. holes B, C and D at Site 846) provided the least disturbed representation of sedimentation.

In this work we continue to use the Match program developed by Lisiecki and Lisiecki (2002), but take advantage of a high resolution downhole log of conductivity acquired at the time of

drilling, which provides the best representation of a truly continuous and undistorted sedimentary column at Site 846.  This opportunity was not recognized in the compositing of Lisiecki and Herbert (2007), although it could have been apparent based on the work of Harris et al. (1995), who used a different HLDT log that has lower spatial resolution than the conductivity sensor log analyzed here.  We use results from the FMS conductivity sensor, which acquired data at 0.25

cm resolution from ~80 mbsf (early Pleistocene) to the base of the section (see http://brg.ldeo.columbia.edu/logdb/hole/?path=odp/leg138/846B/).  Unlike many of the borehole logging measurements, the conductivity log preserves fine scale (centimetric) variations related to changes in lithology and sediment physical properties. The conductivity measurements essentially represent the inverse of GRAPE logging, as conductivity increases with conductive

pore water volume and decreases as the volume of solids grows.  The conductivity log is therefore quite similar to variations in web bulk density logged by the GRAPE sensor in cores raised during drilling of Site 846 (Figure 1 reports highly significant correlations of detrended data). As demonstrated by shipboard results, web bulk density correlates positively with carbonate content and therefore positively with reflectance (r = 0.69 for datasets from Site 846).


The downhole log information includes conductivity measurements from three pad sensors.  We tested different combinations of weighting the log data against reflectance and GRAPE time series measured on cores (mean, geometric mean, maximum/minimum of the 3 measurements) and found that choosing the minimum of the 3 sensor measurements at each depth interval gave

consistently the highest match to the core MST data sets.  We interpret this result to reflect the likelihood that high conductivity readings can result from poor pad contact with the borehole and/or washouts of sediment, and are likely to be outliers.  We also found that one interval of the borehole provided unreliable data from any of the sensors, most likely due to a significant washout, and deleted data from this segment (138.37 to 138.19 meters in log depth) in our

alignment process.  Three more short segments showed unusually low conductivity and were

likewise removed (146.6799-146.746, 150.3299-150.3705, and 150.4315-150.4798 meters log depth). We used core photographs to remove GRAPE and reflectance outliers at the top and base of cores based on visual evidence of core disturbance.

Mapping between offset holes and the downhole conductivity log was done to optimize the alignment of GRAPE and reflectance data to the target. In addition, guided by the downhole log, we optimized the alignment of each offset hole data to the other offset holes. Data sets were normalized and detrended and we mapped (+) reflectance and GRAPE bulk density to (-) conductivity. While we relied on the reflectance and GRAPE information, wherever possible we

checked for consistency with stable isotopic (Shackleton et al., 1995b) and alkenone (Lawrence et al., 2006) data sets as well. Tie points were inserted based on both reflectance and GRAPE data, but the final composites were generated from the Match algorithm applied to the GRAPE time series only, as this consistently showed better alignment to the log and smoother variations in implied core distortion than the reflectance data- this presumably reflects the benefit of

consistent GRAPE calibration to density standards in comparison to the reflectivity measurements.

    Our goal in composite section generation was to achieve alignments between the depth series of offset holes and the conductivity log robust at the scale of orbitally-related variations in

sedimentation and proxy variance. Given sedimentation rates in the interval of interest of 4-5 cm/kyr, this meant achieving satisfactory alignment at ~40 cm, or one half precessional wavelength. In practice, we found that approximately 20 tie points served to achieve this level of match. The Match algorithm includes a number of tradeoffs in the alignment procedure. Essentially, it determines the optimal alignment of segments of data at integer ratios, with a 1:1

alignment indicating no distortion of one record relative to the other. In addition to default choices supplied by the Match software (http://lorraine-lisiecki.com/match.html) we ran an experiment where we increased the number of integer choices in the vicinity of the 1:1 match in order to suppress smalls-scale jumps in relative accumulation in order to see larger patterns between offset holes (the point penalty score for correlations to the log can be improved by about

15% by letting the accumulation rate change more freely in the Match algorithm than we do for the "smooth" mapping). To enhance smoothness, we assigned a relatively high penalty function to the speed parameter but very little penalty to the speed change parameter- this allowed for fine scale adjustment of relative accumulation rates while minimizing large and abrupt changes in the mapping functions. We also iteratively adjusted the gaps assigned between cores from the

nominal gaps determined by the shipboard splices, so that the matching procedure yielded a smooth downhole mapping without artificial jumps at core breaks (Figures 2 and 3.

    Ultimately, we determined that choices such as the number of intervals and penalty parameters had very little effect on the final mapping (e.g. variations in parameters by a factor of 2 produced

negligible changes in mapping). Final Match parameter choices are presented in Table I. Figure 2 portrays the final alignment of individual GRAPE time series to the downhole conductivity log at Site 846. At Site 850, we produced a composite section from simultaneous alignment of GRAPE, reflectance, and magnetic susceptibility measurements at holes 850A and 850B. This composite has no downhole log reference section; however, the composite allows us

to very accurately map discrete measurements between holes A and B of alkenone unsaturation



and stable isotopes produced as part of the present work. Composite reflectance, GRAPE and magnetic susceptibility (for Site 850) sections are included as Supplementary Tables 1-13.

**1.2 Stable isotopic data**

Because our initial investigation suggested an ambiguity in the composite section based on splices of holes 846D (which provides the majority of the backbone to the shipboard composite section) to 846C in the interval 95 to 181 meters composite depth (mcd) as defined by the
shipboard scientific party (splice of 846D core 13 to 846 C core 13 and then to 846D core 14), we analyzed 227 new samples from hole 846B, cores 12, 13, and 14, of which 211 yielded enough benthic foraminifera for isotope analysis at the Brown University stable isotope facility. Samples were freeze-dried, soaked in water for 24 hours to disaggregate, and wet sieved using a 150 μm mesh. Samples were then dried at 40 °C and split into a faunal and isotope fractions.
Isotope fraction vials were dry sieved, benthic foraminifera were picked from 150-355 μm, cleaned with 70 μl ethanol, sonified for 30 seconds, ethanol was drawn off with a pipette, and specimens were dried overnight at room temperature. In some cases, it was necessary to expand the size fraction to 150-355 μm to provide enough carbonate for replicate measurements. Isotope ratios were quantified at with a Finnigan MAT 252 with Carbonate Kiel III autosampler, where
the individual sample is reacted with 70 °C $H_3PO_4$. 170 of the 211 intervals were analyzed at least in duplicate, and in a number of cases we were able to acquire triplicate or quadruplicate replication. We analyzed both *Cibicidoides wuellerstorfi* and *Uvigerina peregrina* (95 paired samples) and determined an average offset of +0.643 ± 0.011 $^o/_{oo}$ (standard error of the mean) for $\delta^{18}O$ (*Uvigerina* heavier) and + 0.973 $^o/_{oo}$ for $\delta^{13}C$ (*Cibicidoides* heavier, ± 0.018 standard
error of the mean). At Site 846, we supplemented analyses with a handful (15) of measurements on *Cibicidoides mundulus* where *C. wuellerstorfi* and/or *U. peregrina* were not abundant enough to provide a good signal for mass spectrometry. In accordance with standard practice, *Cibicidoides* $\delta^{18}O$ values were adjusted to *Uvigerina* values; similarly, *Uvigerina* carbon isotope values were adjusted to *Cibicidoides*. *C. mundulus* isotopic values were taken as equivalent to *C.*
*wuellerstorfi*. Wherever replicate analyses were possible, we report results as the average.

At Site 850, we obtained 273 samples from Holes A (core 7) and B (cores 6 and 7) where benthic foraminifera were sufficient for isotope analysis (we had smaller sample volumes, so the data set is less continuous than for Site 846). Replication is also less extensive (155 of 273 samples) and
the proportion of *C. mundulus* used was higher (63 samples). Isotopic adjustments by species were handled identically to those at Site 846.

When we compared stable isotopic data acquired at Brown with values reported by Shackleton et al. (1995b) over the equivalent interval at holes 846D and C it was evident that there was an
isotope offset of +0.17 in $\delta^{18}O$ (Brown heavier). Variance was essentially identical; therefore, we adjusted the Brown values to align with the much longer Shackleton et al. (1995b) record by adjusting the Brown $\delta^{18}O$ values by the average difference. It is not clear that the choosing the Shackleton values is preferable on analytical grounds, but our choice allows the user to incorporate our new splice without having to adjust previously published values determined by
the Shackleton et al. (1995b) study. We found no statistically significant offset between the carbon isotope values determined at Brown and the values in Shackleton et al. (1995b).





The oxygen and carbon species-specific isotopic offsets we report are identical with ($\delta^{18}O$) or very similar to ($\delta^{13}C$) the offsets used by Shackleton et al. (1995b).


### 1.3 Alkenone unsaturation estimates

Alkenone paleothermometry relies on the temperature dependence of the degree of unsaturation (number of double bonds) observed in the suite of organic compounds ($C_{37:3}$ and $C_{37:2}$ alkenones) synthesized by marine surface-dwelling haptophyte algae (Marlowe et al., 1984; Prahl and Wakeham, 1987). Alkenone extraction followed freeze-drying ~1 g of homogenized dry sediment, using 100% Dichloromethane (DCM) and a Dionex 200 Accelerated Solvent Extractor (ASE. Prior to quantification, extracts were evaporated with nitrogen and reconstituted with 200 µL of toluene spiked with n-hexatriacontane ($C_{36}$) and n-heptatriacontane ($C_{37}$) standards.
Alkenone parameters were determined using an Agilent Technologies 6890 gas chromatograph–flame ionization detector (GC-FID), with Agilent Technologies DB-1 column (60 m, 0.32 mm diameter and 0.10 mm film thickness. Procedure entailed a 1µl injection, initial temperature 90 °C, increased to 255 °C with 40 °C/minute rate, increased by 1 °C/minute to 300 °C, increased by 10 °C/minute to 320 °C, and an isothermal hold at 320 °C for 11 minutes. All GC analyses
simultaneously provide the information to determine the $U^{K'}_{37}$ unsaturation values and an estimate of the total amount of $C_{37}$ ketones by summation of the areas of the $C_{37:2}$ and $C_{37:3}$ alkenones. Long-term laboratory analytical error, estimated from replicate extractions and gas chromatographic analyses of a composite sediment standard is equivalent in temperature to ±0.1 °C.


We analyzed a total of 249 samples from hole 846B and 427 from holes 850A and 850B. All samples yielded adequate alkenone concentrations for unsaturation estimates. The mean $U^{k'}_{37}$ value for the interval of interest from hole 846B of 0.896 compares very well to the average reported by Lawrence et al. (2006) from equivalent interval of hole 846D of 0.902, although
small differences in orbital-scale peak amplitudes are observed that probably reflect small variations in gas chromatographic performance over time. For consistency to the longer Lawrence et al. record, we adjusted new Uk'37 values by +.006. To convert to estimated sea surface temperature (SST) we used the global core top calibration of Muller et al. (1998).

## 2 Results

### 3.1 Lessons on coring distortion

The Lisiecki and Herbert (2007) study of hydraulic piston coring relied on offset holes alone to assess distortion related to porosity rebound and coring-induced disturbance. This meant that distortion could be assessed from one hole *relative* to another, but without a definitive undistorted reference (with the exception of reported drilling depth at the top and bottom of each core). In the present case, we can assess distortions using the Match alignment of composite sections to the presumably undistorted conductivity borehole log. Several conclusions stand out.
First, there are very few instances of coring compression indicated by the Match algorithm (compression would register as a relative accumulation rate < 1) (Figure 3). Second, we can only



document one instance of coring repetition, a segment of ~40 cm at the top of 846B core 16,
consistent with the original composite section proposed by the shipboard party.

Lisiecki and Herbert (2007) found strong correlations of extension for Leg 138 sites with
%CaCO$_3$, and GRAPE density for the upper 50m, but decreasing correlation at depth.  Our
mapping of each offset hole to the downhole log shows a relationship of coring distortion
between the inverse of log conductivity, composite GRAPE density, and composite reflectance
(not shown) over the study interval using the smoother fit option (e.g. more Match speed choices
close to 1:1 between cores and the downhole log) (Figure 3). Most of the distortions correlate
between offset holes (Figure 3), demonstrating the largely predictable coring expansion with
lithological variations.  In the case of these siliceous sediments, the coring distortion may also be
related to lithology as well as physical properties, as zones with lower density (higher porosity)
generally have high biogenic silica contents and may behave differently from carbonate-
dominated lithologies during coring.  While the distortion appears systematic, it somewhat
counter-intuitively indicates expansion of the lower conductivity (higher carbonate/lower
porosity) beds relative to lower carbonate beds when examined in detail, although the
relationship is not statistically significant (Figure 4) and may depend on unresolved factors such
as differences in the proportion of biogenic silica to detrital content in the non-carbonate
fraction, or to the *contrast* of physical properties with depth.  Evaluating distortion as a function
of position within each core fails to reveal systematic distortion as a function of the HPC barrel
penetration (Figure 5).  It therefore seems as if there is little differential distortion during the
stroke of the HPC that cannot be explained by the behavior of contrasting lithologies penetrated.

We also found that our new composites of reflectance and GRAPE density (see Tables S2-S7)
compare much more favorably to the downhole log data than the prior Lisiecki and Herbert
(2007) composite.  We attribute the better match both to the use of a continuous log reference
and to the additional splicing constraints gained from consulting stable isotopic and alkenone
data.  This observation highlights the importance of incorporating all available stratigraphic data
to avoid mistakes in tying potentially ambiguous sections between offset holes so as to generate
the most reliable composite section.

### 3.2 Revision of the Mammoth-Gauss/Gilbert equivalent section
To generate a revised composite, we used the downhole conductivity log as our reference.  The
original splice links holes 846D (the primary hole used by Shackleton et al., 1995b, and followed
by Lawrence et al. 2006) and 846C. As Figure 6 demonstrates, recovery at hole 846C entirely
omits the M2 interval.  We turn to hole 846B, which contains the best representation of both the
M2 interval and the subsequent interglacial recovery, to tie its record to hole 846D.  The original
composite using hole 846C does nearly completely tie the gap between cores 12 and 13 of hole
846B, but is extremely stretched for the last ~2 meters by coring distortion (Figure 6).  The tie
lines indicated in Figure 6 include important constraints from the GRAPE log and from our new
discrete δ$^{18}$O and alkenone measurements (not shown) in addition to the reflectance data
displayed.



The original δ¹⁸O section published by Shackleton et al. (1995b) in the vicinity of the M2 glacial
event seems unusual in having 2 distinctly separated enriched features (Figure 7) - an anomaly
that showed clearly in the original LR04 isotope stack (Figure 9 of Lisiecki and Raymo, 2005),
but that nevertheless guided the final product of that stack. Carbon isotopes (not shown) also
show an anomalous depletion in the same interval. Turning to a high-resolution SST record
previously generated at Site 846 (Lawrence et al., 2006), it too seems anomalous, in that a strong
cooling is very short lived and does not follow the reflectance, GRAPE, and stable isotopic
trends well in this particular interval (Figure 9). As documented below, we are convinced that an
error of unknown origin at Site 846 has confused the stratigraphy surrounding the M2 event.

Both the stable isotopic and alkenone data previously generated at hole 846D deviate in an
anomalous manner relative to reflectance and GRAPE variations in the M2 interval (Figures 7 &
8). In general, the isotopic and alkenone values align well with variations in reflectance, log
conductivity and also GRAPE bulk density at Site 846; this is the only interval we have observed
with such significant deviations between proxies. In contrast, if we exclude the problematic
intervals identified in Figure 9 (specifically, stable isotope data from 846D core 13, section 1,
and alkenone data from 846D core 13, section 2), the new composite that integrates newly
generated stable isotopic and alkenone data from hole 846B follows reflectance and log data
closely through the interval containing the M2 glacial event (Figure 9). It is therefore apparent
that, for unknown reasons, both previously published isotopic and alkenone data that followed
the original splice in the M2 interval are not reliable. We attempted several simple fixes, such as
assuming that stable isotope data from hole 846D, core 13, section 1, were reversed in depth by a
sampling error, but the data still would not align well with the new isotope section from hole
846B, or with the reflectance and GRAPE data generated on the same section at 846D. We
conclude that stable isotopic data from 846D, core 13, section 1, and alkenone data from 846D,
core 13, section 2 must be discarded as erroneous. In addition, the original splice from 846D,
core 12 to 846C, core 12 is problematic because of what appears to be extensive coring extension
at the base of core 12 at Hole 846C (Figure 6).
We can confirm the reliability of the new isotopic and SST composites at Site 846 (see Tables S8
and S9) by comparing them to newly generated records at Site 850 (Figure 10; Table S10). The
equivalent interval at Site 850 spans only one core break, minimizing possible uncertainties in
creating a continuous composite section there. Unfortunately, the Pliocene section at Site 850 is
too shallow to have had borehole logging, so we can only create a composite depth section based
on coring. Nevertheless, isotopic and reconstructed SST patterns can be matched very precisely
(sample tie lines indicated in Figure 10) between Sites 846 and 850, once coring and/or
sedimentation rate distortions are considered. These ties also allow us to indirectly transfer the
high-quality magnetic polarity stratigraphy obtained at Site 850 to its equivalent positions at Site
846 (Figure 10).

## 4 Conclusions

In our new composite, we align data from offset holes to the borehole conductivity log to obtain
the least distorted representation of the mid to late Pliocene interval, focusing especially on the
stratigraphy around the M2 interval. Careful analysis of the original composite section produced
at Site 846 shows errors of unknown origin in the critical interval surrounding the M2 glacial





event. These errors influenced the original LR04 isotopic stack and the composite tropical ocean
temperature stack of Herbert et al. (2010), and persist in the recent Ahn et al. (2017) revision to
LR04, although to a lesser degree as more sites have been incorporated into the new isotopic
stack.  In contrast to the earlier data sets from Site 846, the M2 glaciation now stands out as a
long sawtooth feature of enriched δ$^{18}$O and cold SST values, rather than as 2 events separated by
a significant deglaciation/warming.  The anomalously enriched interval of the δ$^{18}$O record
preceding the M2 glaciation identified as MG4 in LR04 (see Figure 9 of Lisiecki and Raymo,
2005) seems to be a 1-point outlier when new isotopic information from Hole 846B is spliced
into the revised composite (Fig 8). We suggest that the new splice and composite section shown
in Figure 9 replace the previously published alkenone and stable isotope sections of Lawrence et
al. (2006) and Shackleton et al. (1995b) and that this revised section (see Supplementary Tables
S8 and S9) be incorporated in future stable isotope and temperature stacks.  Given the expanded
sedimentation rate and dense sampling resolution of stable isotopic and alkenone data at Site
846, this new composite provides one of the best representations of late Pliocene
paleoceanographic variability.





## Code/Data Availability

Composite sections of non-destructive measurements (reflectance, GRAPE wet bulk density, magnetic susceptibility, borehole conductivity log from Site 846) at Sites 846 and 850 are available, with primary (hole, core, section, and centimeter depth) as well as derived (composite depth) information included. New stable isotopic and alkenone determinations at Sites 846 and 850 with primary and derived (composite depth) information are also reported. All data are

available as tab-delimited text files archived on the website Pangaea.

## Author Contribution

T. Herbert produced composite sections using the Match algorithm and wrote the bulk of the manuscript. All authors contributed to manuscript revision. R. Caballero-Gill produced

alkenone SST estimates at Sites 846 and 850 and helped in the construction of composite sections. J.B. Novak picked foraminiferal samples for stable isotopic analysis and supervised running samples at the Brown Stable Isotope Facility.

## Competing interests

The authors declare that they have no competing interests associated with this manuscript.

## Acknowledgements

N.S.F. grants OCE-1459280 and PIRE-1545859 supported Herbert and Caballero-Gill during this project; stable isotopic analyses and alkenone analyses were also supported by H.J. Dowsett

of the U.S. Geological Survey. The authors gratefully acknowledge the curators of the Integrated Ocean Drilling Program for providing samples from ODP Sites 846 and 850.





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



Table 1: values used with Match algorithm to generate "smooth" mappings of offset holes for
new composite sections and for mapping to the borehole conductivity log. The default mapping
for the "high resolution" mapping used speeds of 1:3, 2:5,1:2, 3:5, 2:3, 3:4, 4:5, 1:1, 5:4, 4:3,3:2,
5:3, 2:1, 5:2, and 3:1

| Match parameter | |
|---|---|
| numintervals | 2000 composites, 4000 to borehole log |
| nomatch penalty | 40 |
| speed penalty | 45 |
| target speed | 1:01 |
| tie penalty | 60 |
| gap penalty | 40 |
| speeds | 10:7 4:3 9:7 5:4 6:5 7:6 8:7 9:8 10:9 11:10 13:12 21:20 31:30 1:1 |
| | 30:31 20:21 12:13  10:11 9:10 8:9 7:8 6:7 5:6 4:5 3:4 7:10 |






Supplementary Tables

Table 1: Borehole conductivity log
Table 2: 846B GRAPE composite
Table 3: 846C GRAPE composite
Table 4: 846D GRAPE composite
Table 5: 846B reflectance composite
Table 6: 846C reflectance composite
Table 7: 846D reflectance composite
Table 8: 846 composite isotopic data
Table 9: 846 composite SST data
Table 10: 850 composite SST and isotopic data
Table 11: 850 composite GRAPE data
Table 12: 850 composite reflectance data
Table 13: 850 composite magnetic susceptibility data



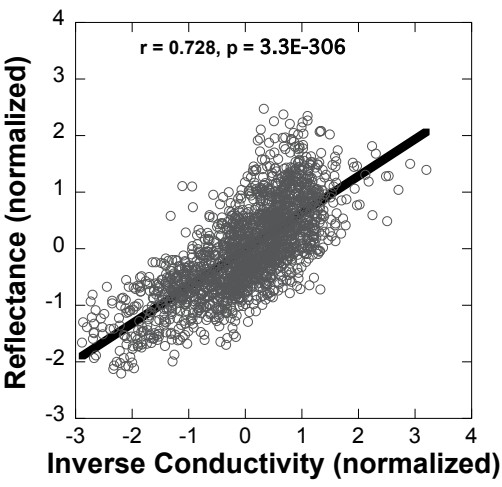

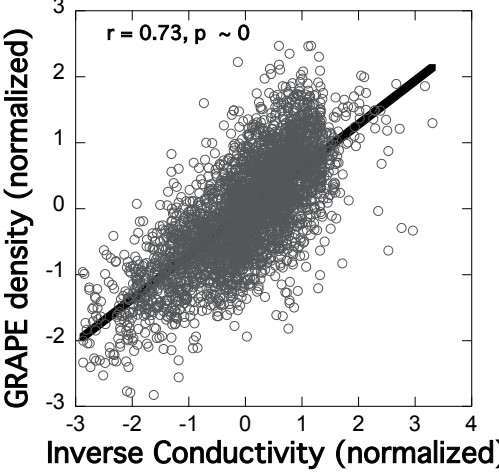

Figure 1. Correlations of the inverse of borehole log conductivity (detrended and normalized) to GRAPE density (detrended and normalized) and to color reflectance (channel 1, detrended and normalized) over the depth interval 85-170 mbsf (95-181 mcd), hole 846B. Note that the inverse of conductivity is plotted. Both correlations (N = 1853, 3388) are highly significant. Correlations of the same variables from holes 846C and 846D were very similar

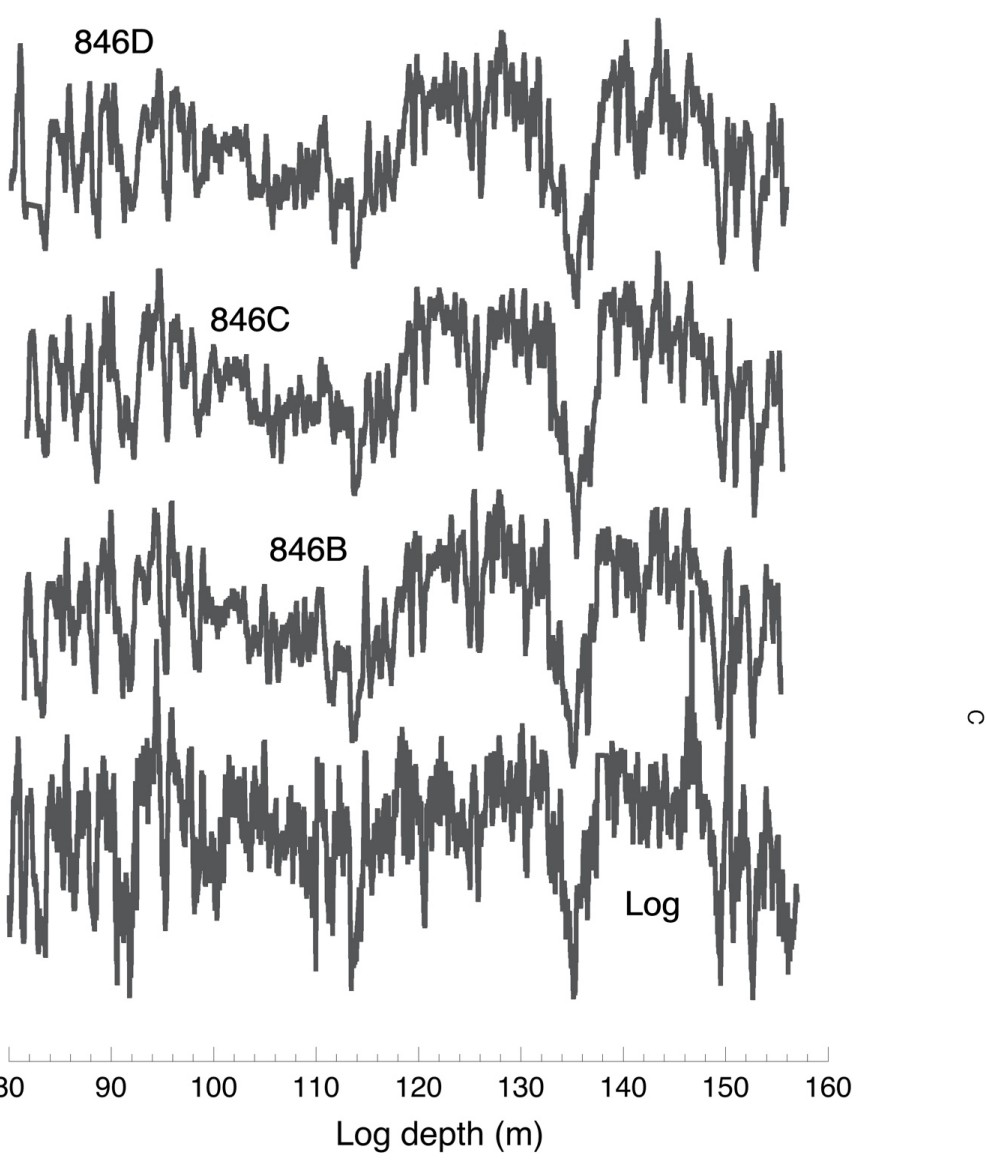

Figure 2. Composite sections of GRAPE density generated using the Match algorithm for holes 846B, C, and D, on a new composite depth section, compared to the inverse conductivity log depth. All data sets were detrended and normalized for this comparison. Coring gaps at each hole have been filled in from offset holes to make continuous splices.

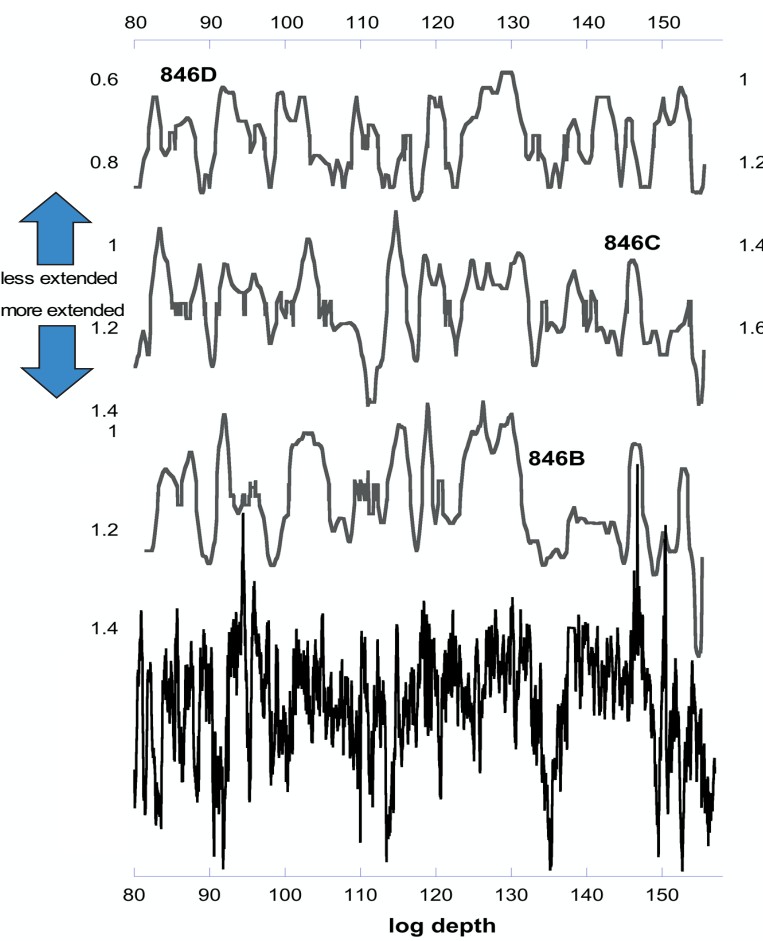

Figure 3. Coring distortion inferred from Match alignment of GRAPE composite sections to the borehole conductivity log (log has been inverted, detrended, and normalized) generated using the "smooth" Match parameters emphasizing a 1:1 depth mapping of core data to the downhole log. Values >1 indicates stretching of the cored section relative to the borehole log, which we assume represents the best representation of the in situ stratigraphy. Note that the scale for coring distortion has been inverted so that higher expansion (stretching) is in the downward direction.






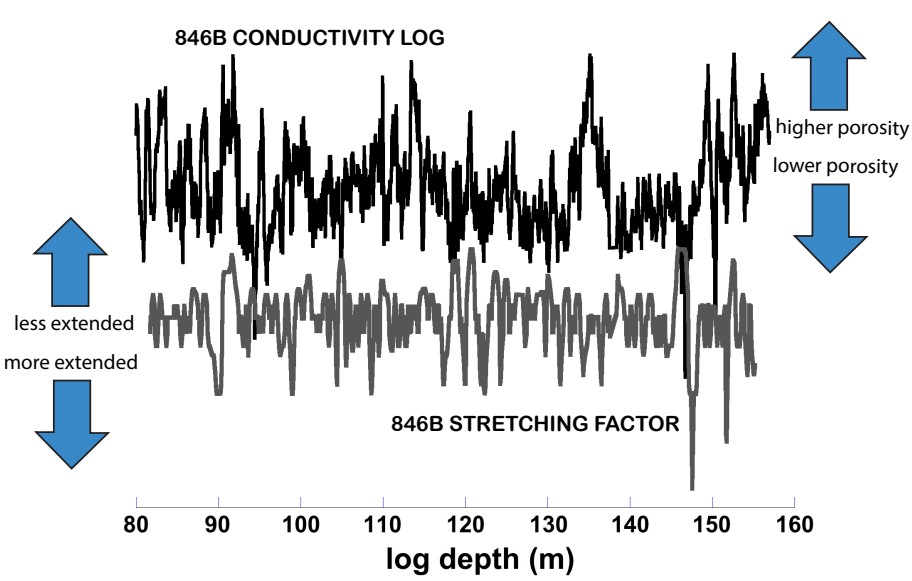

Figure 4. Coring distortion inferred from Match alignment of the hole 846B GRAPE composite section to the borehole conductivity log (conductivity data detrended and normalized) generated using the "high resolution" Match parameters emphasizing the closest possible mapping of core data to the downhole log. Note that core stretching generally moves positively with lower conductivity/porosity (higher carbonate content).





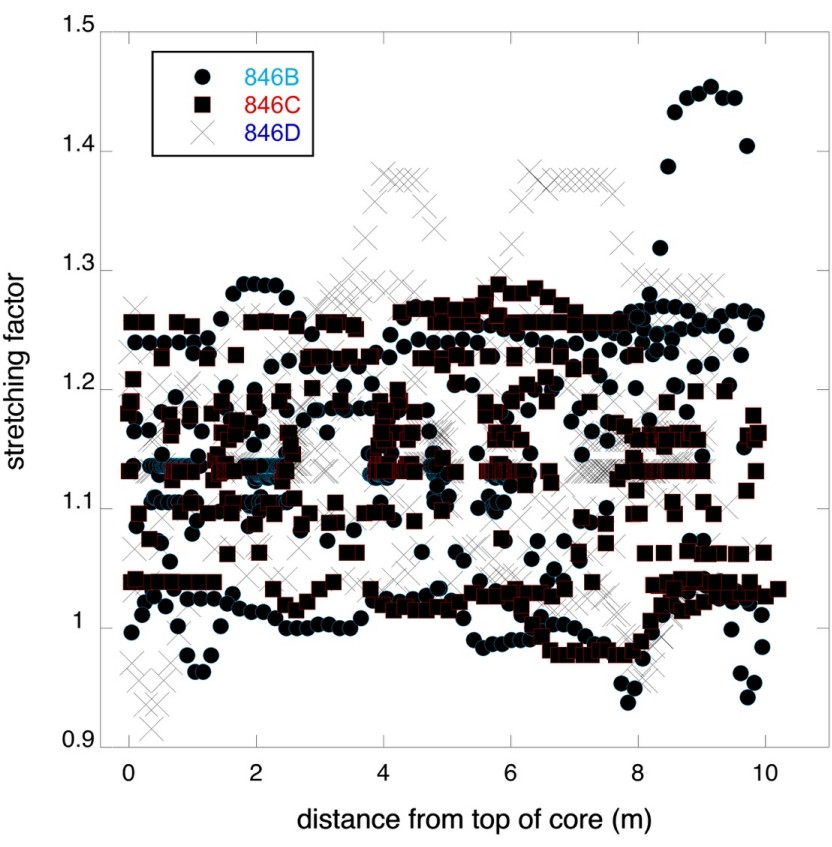

Figure 5. Inferred coring distortion based on Match alignment of GRAPE composite sections to the borehole conductivity log as a function of depth in each ~9.5 m core. Note that there is no evidence for a systematic pattern of stretching/squeezing as a function of penetration depth.



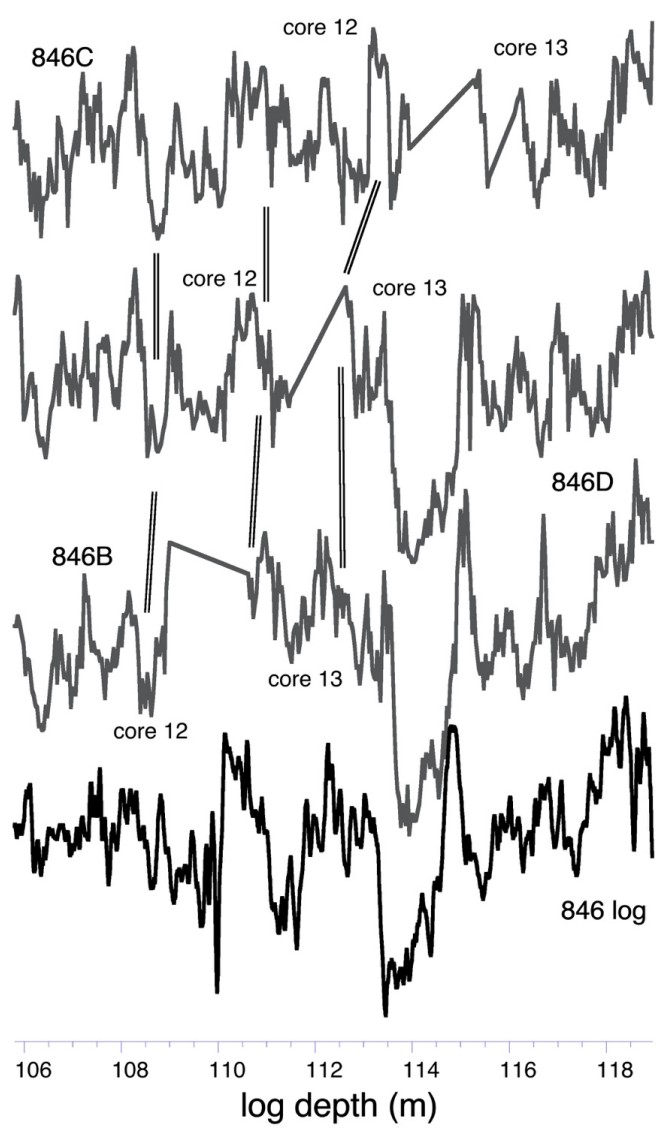

Figure 6. Detail of reflectance-based correlations (original mcd 122.6-137.8) of offset holes 846B, 846C, and 846D correlated to the common depth scale of the borehole log (log depth 106-119 m). Signals were detrended and normalized; higher reflectance and lower conductivity plot upward. Note the problematic tie of the base of Hole 846C, core 12 to the top of Hole 846D, core 13 (shipboard splice). In contrast, Hole 846B has a sequence that correlates easily to the borehole log and can be reliably spliced to the base of Hole 846D, core 12.

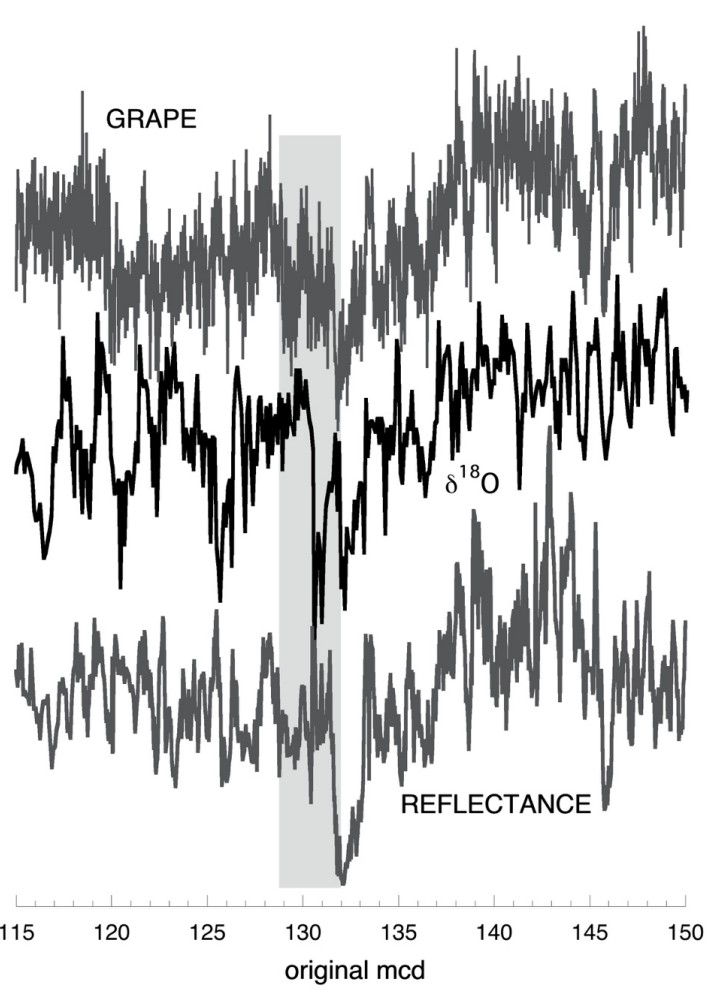

Figure 7. Comparison of $\delta^{18}O$ series of Shackleton et al. (1995b) on original composite depth (Shipboard Scientific Party, 1992) in comparison to reflectance and GRAPE data (original shipboard composite. Note the anomalous interval indicated by gray shading where the isotopically enriched values depart from reflectance and GRAPE values.

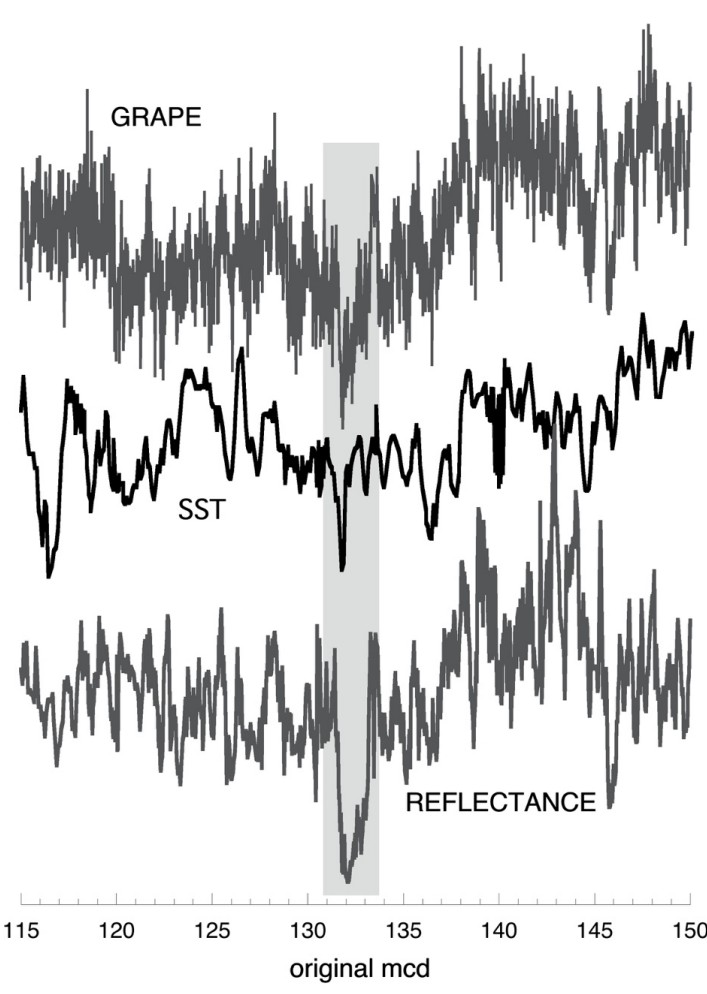

Figure 8. Comparison of alkenone-based SST series of Lawrence et al. (2006) on original
composite depth in comparison to detrended and normalized reflectance and GRAPE data. Note
the anomalous interval indicated by gray shading where the warm SST values depart from
reflectance and GRAPE values.





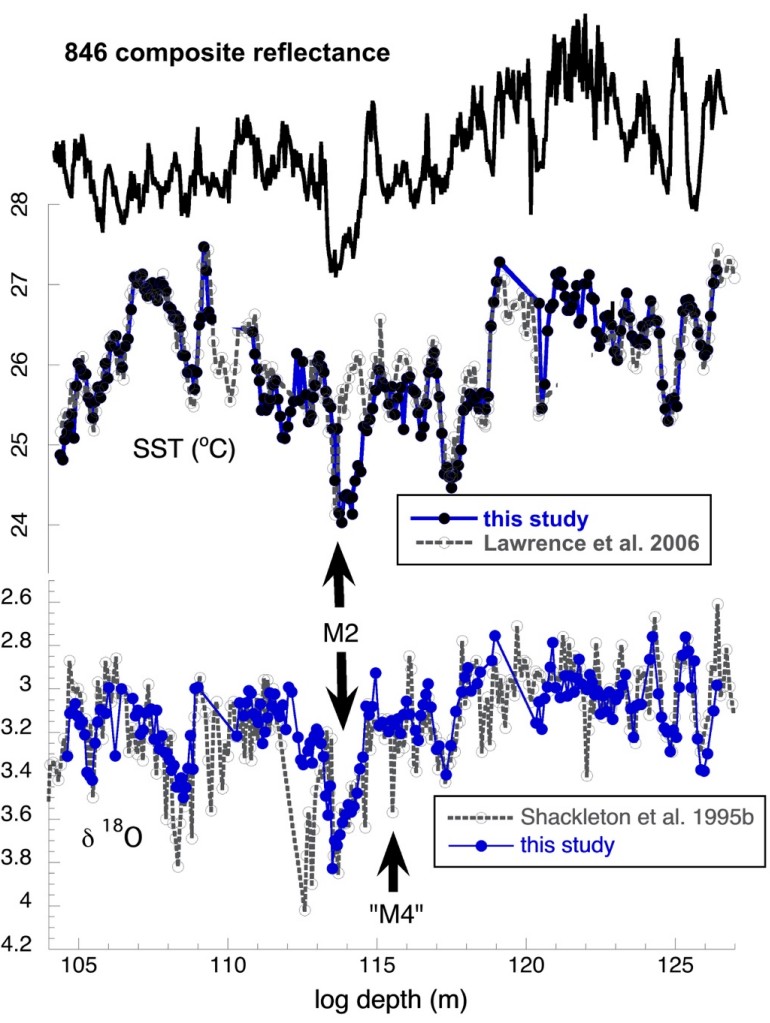

Figure 9. Revised alkenone and δ18O time series based on new data from hole 846B, aligned with prior data from Shackleton et al. (1995b) and Lawrence et al. (2006), excluding the problematic intervals around the M2 glaciation (see text. The composite section has been mapped to the borehole log depth. Previously published isotopic and alkenone data in the problematic M2 interval are shown as dashed lines. Revised composite time series now align well, and the interval containing the M2 glacial event presents as a sawtooth isotopic enrichment and cooling.

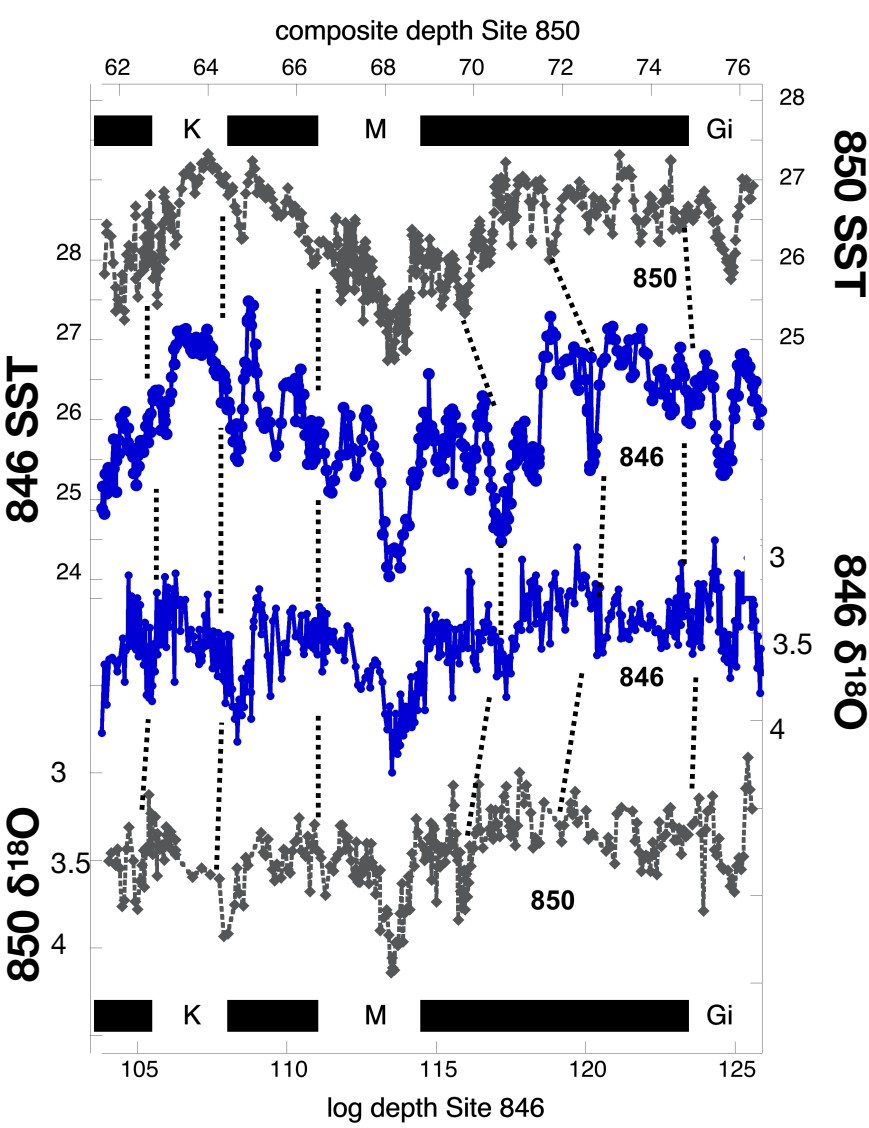

Figure 10. Comparison of the new alkenone-based SST and δ18O composites from Site 846 to
composites from Site 850. The location of magnetic polarity reversals (reversed Kaena,
Mammoth and Gilbert polarity zones indicated by abbreviations) at Site 850 can be mapped to
Site 846 with high confidence by the close correspondence of SST patterns (note: polarity
determinations are only available at Site 850).