# Peer review of "A Revised Mid-Pliocene Composite Section Centered on the M2 Glacial Event for ODP Site 846"

_Climate of the Past, 2019_

## Referee Comment (RC1) · Mitch Lyle (Referee) · 3 Mar 2020

Herbert et al noted that the Site 846 composite section across the Pliocene M2 glaciation was anomalous compared to other sites. They attempt to remake a composite section for ODP Site 846 based on physical properties from shipboard scanning tracks but also by correlation to the wireline logs, and comparison to new discrete stable isotope and alkenone analyses. It has long been known that one hole, even if completely cored, will have gaps between cores taken. Two holes are better, but there can still be gaps. There are times when correlations are ambiguous. They develop a methodology to compare shipboard data to a conductivity record from the wireline logs and check it with other discrete measurements.

They used a process where they compared a high-resolution electrical conductivity log to the shipboard bulk density measurements, a measure of porosity. In unconsolidated marine sediments, conductivity is directly related to the seawater present in the measurement volume, or porosity. They report that the record is from the HLDT tool, but this is incorrect. Comparing the data table to the archived logs, they are using conductivity data from the Formation MicroScanner, an imaging conductivity tool, on pass 2 of Hole 846B. It is important to keep the correct description of the data and they should make note of where it is archived (http://brg.ldeo.columbia.edu/logdb/hole/?path=odp/leg138/846B/). The detrending and normalization of the data are warranted, since the FMS is not well calibrated to absolute values.

Elimination of flyers in both the logging and GRAPE data is justified and well explained.

Lines 258-274: I was impressed that the coring distortions were correlated across holes and apparently result from different responses by different lithologies to the coring. The most extension was in the more porous diatom-rich sediments, as might be expected.

Figure 6 and Fig 7: some scales on the graph would be useful. On reflectance, is white up or down? Is 18O reversed?

I noticed that there was no table that describes the revised composite section. An interval table that shows their new proposed composite is needed for other researchers. It isn't clear from the text exactly what holes are involved, and where different cores are joined. They would only have to do the interval they have revised, e.g, like this section of Site 849 where each line represents an interval from each hole included in the composite.

| Site_Top | Hole_Top | Core_Top | Type_Top | Section_Top | Interval_Top | CSF_Top | CCSF_Top | Section_Bottom | Interval_Bottom | CSF_Bottom | CCSF_Bottom | TIE    |  |
|----------|----------|----------|----------|-------------|--------------|---------|----------|----------------|-----------------|------------|-------------|--------|--|
| 043      | U        | 14       | ^        | 1           | . 03         | 102.33  | 124.03   | 3              | 103             | 112.33     | 121.73      | 11E    |  |
| 849      | В        | 13       | н        | 1           | . 64         | 111.84  | 127.94   | 6              | 119             | 119.89     | 135.99      | TIE    |  |
| 849      | D        | 13       | х        | 3           | 32           | 118.82  | 135.98   | 5              | 100             | 122.5      | 139.66      | TIE    |  |
| 849      | В        | 14       | x        | 1           | . 86         | 121.56  | 139.66   | 3              | 129             | 124.99     | 143.09      | TIE    |  |
| 849      | D        | 14       | x        | 1           | . 1          | 125.11  | 143.26   | 4              | 136             | 130.96     | 149.11      | APPEND |  |
| 849      | В        | 15       | x        | 1           | . 66         | 130.96  | 149.11   | 7              | 3               | 139.33     | 157.48      | TIE    |  |
| 849      | D        | 15       | x        | 2           | 8            | 136.38  | 157.48   | 6              | 78              | 143.08     | 164.18      | TIE    |  |
| 849      | В        | 16       | x        | 3           | 49           | 143.49  | 164.18   | 5              | 109             | 147.09     | 167.78      | TIE    |  |
| 849      | D        | 16       | x        | 1           | . 44         | 144.94  | 167.78   | 5              | 54              | 151.04     | 173.88      | TIE    |  |
|          |          |          |          |             |              |         |          |                |                 |            |             |        |  |

---

## Referee Comment (RC2) · Anonymous Referee #2 · 29 Dec 2020

Herbert et al. present a revised composite section during the Mid-Pliocene for tropical East Pacific ODP Site 846. Their new composite section is based on an existing high resolution electrical conductivity log (not previously used to establish a composite section). The authors check their composite section with new benthic stable oxygen, carbonate isotope, and alkenone-derived SST data from the same Site and from Site 850 for comparison.

The presented study is interesting and important as Site 846 is one key record for reconstructing the Pliocene climate of the tropical East Pacific Ocean. The new composite section significantly changes previous interpretations of climatic changes around the well-known glacial M2 event. The claims of the authors are supported by the presented data and the manuscript is well written. As such the manuscript is suitable for

this journal, however, before final publication I have some minor comments that need to be addressed:

I think the authors could better explain why using the conductivity record is the best method to achieve a reliable composite section. For instance, although the authors claim this is true in lines 93-96, there are missing some supporting references.

At most figures, scales are missing for the y-axes. It would also be useful if the authors would label the presented records with a) b) etc. in all figures. This would make it much easier to identify each data record.

The authors use at many locations in the text abbreviations like PRISM, HLDT, ODP, FMS, MST, GRAPE etc. which are not explained at their first appearance.

More detailed comments:

Title: The authors focus on changes around the glacial M2 event. Why not mention it in the title?

Line 15: The authors should specify the "errors in primary data sets"

Lines 60-63: The authors could insert a figure with a map showing the locations of sites 846 and 850.

Line 106: Please change "web bulk density" to "wet bulk density" and elsewhere.

Lines 145-151: The sentence is too long. Please split.

Lines 192-194: The authors should specify "In accordance with standard practice". Did they adjust the values according to their reported offsets or according to other studies? and if according to other studies they should specify to which studies.

Line 321: Can "for unknown reasons" be better specified?

Lines 353-356: This is a new observation and has to be first mentioned in the results/discussion part.

Figures:

Figure 6: The authors should clearly indicate which record is from which hole. It would also be nice to see the sections indicated.

Figure 9: Please change "M4" to "MG4". Please also indicate the cores and sections around glacial M2.

Figure caption 4: I cannot see a clear correlation between the stretching factor and the conductivity log.

―――――――――――――――――

---

## Editor Comment (EC1) · Arne Winguth (Editor) · 6 Jan 2021

Dear Dr Herbert,

Thank you for submitting your manuscript entitled "A Revised Mid-Pliocene Composite Section for ODP Site 846" [Paper #cp-2019-133] to Climate of the Past. I have now received an assessment of the two reviewers, as well as your response to the reviewers concerns.

In summary I consent with the reviewers suggest minor revision based on following issues:

1. Reviewer #1 states that in unconsolidated marine sediments, conductivity is directly related to the seawater present in the measurement volume, or porosity. The reviewer

notes that the record is from the HLDT tool which appears to be incorrect.

2. Reviewer #2 recommends to better explain why using the conductivity record is the best method to achieve a reliable composite section. In addition, It would also be useful if the authors would label the presented records with a) b) etc. in all figures. This would make it much easier to identify each data record.

The feedback provided in the reviewer assessments of your manuscript is important and should be taken into account as you complete your revision. Please include a marked-up manuscript version showing the changes made in your revision. I encourage you to submit a suitably revised version of your manuscript by March 1, 2021.

Sincerely, Arne Winguth Editor

---

## Author Comment (AC1) · 9 Feb 2021

Herbert et al noted that the Site 846 composite section across the Pliocene M2 glaciation was anomalous compared to other sites. They attempt to remake a composite section for ODP Site 846 based on physical properties from shipboard scanning tracks but also by correlation to the wireline logs, and comparison to new discrete stable isotope and alkenone analyses. It has long been known that one hole, even if completely cored, will have gaps between cores taken. Two holes are better, but there can still be gaps. There are times when correlations are ambiguous. They develop a methodology to compare shipboard data to a conductivity record from the wireline logs and check it with other discrete measurements.

[Figure]

They used a process where they compared a high-resolution electrical conductivity log to the shipboard bulk density measurements, a measure of porosity. In unconsolidated marine sediments, conductivity is directly related to the seawater present in the measurement volume, or porosity. They report that the record is from the HLDT tool, but this is incorrect. Comparing the data table to the archived logs, they are using conductivity data from the Formation MicroScanner, an imaging conductivity tool, on pass 2 of Hole 846B. It is important to keep the correct description of the data and they should make note of where it is archived (http://brg.ldeo.columbia.edu/logdb/hole/?path=odp/leg138/846B/). The detrending and normalization of the data are warranted, since the FMS is not well calibrated to absolute values. ——————————————————————— I think the reviewer is confused. We explicitly report data using the FMS log; the Harris et al. paper referenced used the HLDT tool instead. The URL cited in our text is identical to the one referenced above and was already included. We do now note that the data come from pass 2 of the FMS logging operation. ——————————————————————- Elimination of flyers in both the logging and GRAPE data is justified and well explained.

Lines 258-274: I was impressed that the coring distortions were correlated across holes and apparently result from different responses by different lithologies to the coring. The most extension was in the more porous diatom-rich sediments, as might be expected.

Figure 6 and Fig 7: some scales on the graph would be useful. On reflectance, is white up or down? Is 18O reversed? —————————————————————— We agree that this was an omission in the first draft and have clarified the sign of the Y axes —————————————————————————- I noticed that there was no table that describes the revised composite section. An interval table that shows their new proposed composite is needed for other researchers. It isn't clear from the text exactly what holes are involved, and where different cores are joined. They would only have to do the interval they have revised, e.g, like this section of Site 849 where each line represents an

interval from each hole included in the composite. ————————————————————
————- This is a little tricky, since the Match code that we use generates a continuous mapping function rather than a constant offset. The Match mapping is more realistic, but also difficult to summarize in a table. We now provide a table of the typical ODP form (e.g. splice points and revised composite depth section) and revise the text to emphasize how the data tables of GRAPE, reflectance etc report values mapped to log depth as the common depth scale ————————————————————————-

---

## Author Comment (AC2) · 9 Feb 2021

Herbert et al. present a revised composite section during the Mid-Pliocene for tropical East Pacific ODP Site 846. Their new composite section is based on an existing high resolution electrical conductivity log (not previously used to establish a composite section). The authors check their composite section with new benthic stable oxygen, carbonate isotope, and alkenone-derived SST data from the same Site and from Site 850 for comparison. The presented study is interesting and important as Site 846 is one key record for reconstructing the Pliocene climate of the tropical East Pacific Ocean. The new com- posite section significantly changes previous interpretations of climatic changes around the well-known glacial M2 event. The claims of the authors are supported by the pre- sented data and the manuscript is well written. As such the

manuscript is suitable for C1

this journal, however, before final publication I have some minor comments that need to be addressed: I think the authors could better explain why using the conductivity record is the best method to achieve a reliable composite section. For instance, although the authors claim this is true in lines 93-96, there are missing some supporting references.

At most figures, scales are missing for the y-axes. It would also be useful if the authors would label the presented records with a) b) etc. in all figures. This would make it much easier to identify each data record. The authors use at many locations in the text abbreviations like PRISM, HLDT, ODP, FMS, MST, GRAPE etc. which are not explained at their first appearance.

============================ Thank you for this suggestion; adopted ============================

More detailed comments: Title: The authors focus on changes around the glacial M2 event. Why not mention it in the title? ============================= Thank you for this very useful suggestion- adopted in the revised version ============================

Line 15: The authors should specify the "errors in primary data sets" ============================ We don't know the reason why alkenone and isotope data from one core section containing the M2 interval occurred previously. We have clarified that the data appear to be outliers in light of newly obtained information.

Lines 60-63: The authors could insert a figure with a map showing the locations of sites 846 and 850. ====================== Agreed and now map incorporated ============================

Line 106: Please change "web bulk density" to "wet bulk density" and elsewhere. ============================ Apologies for the typos! ============================

Lines 145-151: The sentence is too long. Please split. =========================== revised as suggested ===========================

Lines 192-194: The authors should specify "In accordance with standard practice". Did they adjust the values according to their reported offsets or according to other studies? and if according to other studies they should specify to which studies. =========================== The classic Shackleton and Opdyke (1973) and Duplessy et al. (1984) papers, whose isotopic offsets we use, are now cited ===========================

Line 321: Can "for unknown reasons" be better specified? =========================== We wish... The samples were run by two different investigators and we simply don't know how to explain the apparent erroneous results. We suggest in the text that samples may have been mislabeled or the core section inverted during sampling. ===========================

Lines 353-356: This is a new observation and has to be first mentioned in the results/discussion part. =========================== For a short paper of this type, we prefer to keep the text as is- we feel that the focus on improving the stratigraphy around the M2 glacial interval works more effectively when all the information and interpretation are presented together.
* * *

---

## Author Response (AR1)

The current version adopts the suggestion of the editor (and reviewers) to define acronyms such as PRISM, HLDT, GRAPE, FMS that may not be familiar to readers of *Climate of the Past*.